# Effects of Exercise-Based Rehabilitation on Lumbar Degenerative Disc Disease: A Systematic Review

**DOI:** 10.3390/healthcare13151938

**Published:** 2025-08-07

**Authors:** Shirin Aali, Farhad Rezazadeh, Fariborz Imani, Mahsa Nabati Sefidekhan, Georgian Badicu, Luca Poli, Francesco Fischetti, Stefania Cataldi, Gianpiero Greco

**Affiliations:** 1Department of Physical Education, Farhangian University, P.O. Box 889-14665, Tehran 19396-14464, Iran; sh.aali@cfu.ac.ir; 2Department of Sports Biomechanics, Faculty of Educational Sciences and Psychology, University of Mohaghegh Ardabili, Ardabil 56199-11367, Iran; rezazadeh.farhad@uma.ac.ir (F.R.); fariborzimani@uma.ac.ir (F.I.); mahsaaanabatiii@gmail.com (M.N.S.); 3Department of Physical Education and Special Motricity, Faculty of Physical Education and Mountain Sports, Transilvania University of Braşov, 500068 Braşov, Romania; georgian.badicu@unitbv.ro; 4Department of Translational Biomedicine and Neuroscience (DiBraiN), University of Study of Bari, 70124 Bari, Italy; gianpiero.greco@uniba.it; 5Department of Education and Sport Sciences, Pegaso Telematic University, 80143 Naples, Italy; stefania.cataldi@unipegaso.it

**Keywords:** corrective exercises, lumbar DDD, therapeutic exercise, physical therapy

## Abstract

**Background:** This systematic review evaluates the efficacy of rehabilitation-focused exercise interventions for lumbar degenerative disc disease (DDD), a leading cause of chronic low back pain. **Methods:** Following PRISMA guidelines, a comprehensive search was conducted across international and regional databases (PubMed, Scopus, Web of Science, Magiran, SID, and Noormags) covering the period from January 2010 to January 2025. The review protocol was registered with the International Prospective Register of Systematic Reviews (PROSPERO) under registration number CRD420251088811. Using keywords such as “lumbar DDD,” “exercise therapy,” and “rehabilitation,” a total of 2495 records were identified. After screening, 20 studies—including clinical trials, quasi-experimental, and experimental designs—met the inclusion criteria and were assessed using the McMaster Critical Review Form for Quantitative Studies. **Results:** Interventions such as hydrotherapy, core stability training, Pilates, and suspension exercises were found to significantly reduce pain and improve functional outcomes. While multimodal approaches (e.g., aquatic exercise combined with acupuncture) showed positive effects, the comparative studies revealed no significant differences between modalities. Suspension training demonstrated superior efficacy in pain reduction compared to isolated core stability exercises. The methodological quality of included studies ranged from good to excellent, with the majority rated as very good or excellent (McMaster scores: 8 “excellent,” 7 “very good,” and 5 “good”). Common limitations among the studies included methodological heterogeneity, small sample sizes (n = 14–30), and insufficient long-term follow-up. **Conclusions:** Exercise-based rehabilitation is an effective strategy for managing lumbar DDD. Evidence particularly supports the use of suspension training and aquatic therapy for superior improvements in pain and functional outcomes. Future research should aim to adopt standardized protocols, recruit larger sample sizes, and include extended follow-up periods to produce more robust and generalizable findings.

## 1. Introduction

Low back pain (LBP) is a multifactorial clinical syndrome characterized by pain, stiffness, or muscular tension localized between the lower rib margin and the gluteal folds, with or without radiating sciatic symptoms [1]. As the leading global cause of disability, LBP affected approximately 619 million people worldwide in 2020, with projections estimating an increase to 843 million by 2050, driven primarily by aging populations and increasingly sedentary lifestyles [1].

The World Health Organization’s Global Burden of Disease Study consistently ranks LBP as the top contributor to years lived with disability (YLDs), exceeding the burden of all other musculoskeletal conditions combined [1,2]. The financial burden of LBP is considerable, accounting for 1–2% of the gross domestic product in developed countries through both direct costs (e.g., imaging, pharmacotherapy, surgery) and indirect costs (e.g., productivity loss, disability claims). In the United States alone, annual expenditures exceed $100 billion, with nearly two-thirds attributed to indirect workplace-related costs [3]. This economic burden disproportionately affects blue-collar workers and middle-aged adults, posing significant challenges for the labor force [2,3].

According to research findings, degenerative disc disease (DDD), encompassing intervertebral disc disorders, has a striking prevalence of 54% and is the leading cause of chronic back pain [4]. This prevalent neuromuscular disorder primarily affects the lumbar and cervical regions of the spine, typically presenting with a triad of debilitating symptoms: localized or radiating pain, muscle weakness, and restricted range of motion [4]. Among its multifactorial causes, persistent pain remains the primary contributor to both temporary and permanent occupational disability [4]. Supporting this, a comparative study by Radziszewski reported work productivity losses exceeding 40% among patients with chronic lumbar DDD. These findings collectively establish comprehensive pain management as a therapeutic cornerstone, which, when combined with functional restoration, serves as the primary indicator of successful treatment outcomes [5]. Extensive research indicates that 70–85% of individuals experience LBP at least once during their lifetime, a prevalence shaped by lifestyle, occupational demands, ageing, and genetic predisposition [6]. Among these cases, nearly 5% progress to disc herniation, in which intervertebral disc bulging becomes a primary source of pain and core muscle dysfunction [7,8]. Although the exact pathological mechanisms remain incompletely understood, several modifiable risk factors—particularly poor posture, inadequate seating ergonomics, and prolonged sedentary behavior—are known to accelerate both disc degeneration and atrophy of core stabilizing muscles [9,10]. These factors increase mechanical stress on intervertebral discs, leading to structural alterations and impaired disc metabolism. The lumbar spine is especially vulnerable, with the majority of herniation occurring at the L4–L5 and L5–S1 levels [11,12]. Pathologically, DDD results from either acute rupture and protrusion of intervertebral disc material or progressive deterioration of surrounding ligamentous and supportive tissues [13]. As demonstrated by the aforementioned clinical cases, the spine’s critical role as the body’s central skeletal axis—mediating the biomechanical relationship between the upper and lower limbs, has a profound impact on overall health and optimal physical function [14,15]. The peak incidence of intervertebral disc herniation occurs most frequently between the ages of 30 and 50, representing the most vulnerable demographic [16]. Clinically, this condition typically presents as radiating back pain that characteristically extends to the knees, hips, and legs, with symptoms notably exacerbated during Valsalva maneuvers such as sneezing and coughing [17]. Anatomical studies consistently identify the L4–L5 and L5–S1 intervertebral discs as the most common sites of pathological involvement, accounting for the majority of DDD cases [18].

For patients with chronic lumbar DDD, strengthening spinal muscles through supervised physiotherapy and well-designed exercise regimens is essential to mitigate deformity progression [19]. Contemporary rehabilitation paradigms reflect an evolved understanding of exercise-related outcomes. While earlier clinical studies advocated generalized exercise regimens, most notably Williams’ flexion exercises, for managing low back pain, more recent biomechanical analyses have demonstrated that such flexion-based protocols may inadvertently increase intervertebral disc pressure, potentially exacerbating the spinal load and compromising structural integrity. This evidence has contributed to a decline in the clinical use of Williams’ protocols, with current practice favoring isometric alternatives [20]. Three evidence-based approaches currently dominate clinical practice: (1) static core stabilization targeting deep segmental muscles [21]; (2) integrated strength and endurance training to restore spinal function [22,23]; and (3) individualized protocols tailored to patient-specific pathology.

Rehabilitation programs are essential for both prevention and management of LBP and require individualised planning and close supervision by trained specialists. Optimal outcomes depend on tailoring exercise selection and program design to the specific DDD subtype (e.g., classified by location, such as cervical or lumbar, degeneration patterns like disc space narrowing, or the presence of features such as osteophyte formation) and the individual symptom profile. Evidence suggests that individualized rehabilitation not only alleviates current symptoms but also reduces recurrence, thereby improving quality of life [14,24]. To address gaps in current rehabilitation practices, this systematic review evaluates the efficacy of exercise-based interventions for lumbar DDD, with an emphasis on identifying strategies for pain relief and secondary prevention.

## 2. Materials and Methods

This systematic review, conducted in 2025, evaluates the therapeutic effects of various exercise interventions within a rehabilitation framework for lumbar DDD.

Following established methodological standards, we utilised the Preferred Reporting Items for Systematic Reviews and Meta-Analyses (PRISMA) guidelines to conduct this comprehensive review [25]. Our systematic process adhered to the standard PRISMA framework, comprising four key phases: (1) initial article identification through database searches, (2) rigorous screening of potential studies, (3) detailed eligibility assessment, and (4) final inclusion of qualifying articles. The full flow of this selection process, including the number of studies identified and excluded at each stage, is illustrated in the PRISMA diagram (Figure 1). The review protocol was registered with the International Prospective Register of Systematic Reviews (PROSPERO) under registration number CRD420251088811.

### 2.1. Search Strategy

A comprehensive systematic search was conducted across six electronic databases: three international (PubMed, Web of Science, Scopus) and three regional Persian-language databases (Magiran, Noormags, and SID).

Additionally, the Google Scholar search engine was queried to ensure thorough coverage of grey literature. The search covered the period from 1 January 2010 to 1 January 2025. Keywords were selected based on established MeSH terms, a review of relevant literature, and pilot searches to optimize sensitivity. The search terms were organized into two main concepts: (1) the population (i.e., lumbar degenerative disc disease) and (2) the intervention (i.e., exercise-based rehabilitation). The intervention keywords were intentionally expanded to include a broad range of modalities such as Pilates, yoga, aquatic therapy, and suspension training to ensure a comprehensive search.

The Boolean operators “AND” (to combine the two main concepts) and “OR” (to combine keywords within each concept) were used. This logical framework was systematically adapted to meet the specific syntax requirements of each database, ensuring a consistent and comprehensive search. For full transparency and reproducibility, the complete search strings used for PubMed, Scopus, Web of Science, and Google Scholar are provided in Appendix A. Equivalent Persian-language keywords were applied for the regional Persian databases. To further ensure comprehensive literature coverage, the reference lists of all included articles were also manually screened using a snowballing method.

### 2.2. Eligibility Criteria and Study Selection

Inclusion criteria were established prior to data extraction to guide the systematic review process. The primary selection criteria focused on studies evaluating the effects of various exercise-based rehabilitation interventions for lumbar DDD. Eligible studies met the following requirements: (1) full-text availability in English or Persian, (2) clinical trial design, and (3) inclusion of human participants to assess the practical efficacy of exercise interventions. Exclusion criteria eliminated studies addressing acute low back pain, specific etiologies of LBP, and those investigating combined treatment protocols (e.g., exercise plus pharmacological therapy) for nonspecific chronic LBP. Studies involving comorbid conditions—such as multiple sclerosis or pregnancy—were also excluded.

The initial screening of titles and abstracts, followed by full-text assessment, was conducted independently by two reviewers (S.A. and F.R.). Any disagreements were resolved by consensus or, if necessary, through consultation with a third reviewer (F.I.). A rigorous screening process was applied after implementing the inclusion and exclusion criteria. The included studies were categorized into two main groups: (1) conventional rehabilitation methods and (2) specialized sports-oriented approaches. These studies were analyzed based on intervention types, exercise outcomes, expert perspectives, muscle electromyographic (EMG) activity, and instrumentation, with findings presented through both textual descriptions and tables.

Participant groups across studies included athletes and non-athletes, with no restrictions on age or gender. Outcomes examined included the effects of suspension training on EMG activity, plantar pressure distribution patterns, and comparisons of diagnostic methods in clinical settings.

### 2.3. Data Extraction

A standardized data extraction form was developed to systematically collect relevant information from all included studies. Two review authors (F.I. and S.A.), both with extensive experience in rehabilitation and systematic review methodologies, independently extracted the data. The extracted information included: (1) study characteristics (first author, year of publication, study design); (2) participant details (sample size, age, gender); (3) intervention characteristics (type, duration, frequency); and (4) primary outcome measures and key quantitative results. Any disagreements between the two reviewers were initially resolved through discussion to reach consensus. If consensus could not be achieved, a third senior reviewer (F.R.) was consulted to arbitrate and make the final decision.

### 2.4. Quality Appraisal

The methodological quality of the included studies was independently assessed by two authors (F.I. and M.N.S.) using the McMaster Critical Review Form for Quantitative Studies [26]. This standardized tool comprises 17 distinct criteria designed to systematically evaluate key aspects of research methodology. Specifically, it assesses the clarity of study objectives, appropriateness of the research design, sampling rigor, validity of outcome measures, intervention protocols, statistical analyses, and the validity of conclusions.

The quality assessment followed the standardised McMaster Review Guide, specifically developed for quantitative research (Table 1). Each item was scored dichotomously: a score of “1” indicated full adherence to the criterion, while a score of “0” reflected non-fulfillment.

Total scores were summed to determine methodological quality and categorized as follows: Poor (0–8), Average (9–10), Good (11–12), Very Good (13–14), and Excellent (15–16).

### 2.5. Data Synthesis and Analysis

Due to significant heterogeneity among the included studies in terms of intervention protocols and outcome measures, a meta-analysis was deemed inappropriate. This heterogeneity was evident across multiple domains: interventions varied widely, ranging from manual therapies (e.g., massage) and mind–body exercises (e.g., yoga, Pilates) to facility-based programs (e.g., aquatic therapy, suspension training).

Furthermore, the studies utilised a wide array of outcome measures, including the Visual Analogue Scale (VAS), the Oswestry Disability Index, and various biomechanical tests, which prevented statistical pooling of the results. Finally, the intervention protocols varied significantly in duration and frequency. Given this high level of clinical and methodological variability, a narrative synthesis was conducted to summarize and describe the findings.

## 3. Results

### 3.1. Study Selection

Our systematic search initially identified 2495 potentially relevant articles through keyword searches. After removing duplicates, 1847 articles remained. Following title and abstract screening, 1720 studies were excluded. The full texts of the remaining 127 articles were assessed for eligibility, which resulted in the exclusion of a further 107 studies. The primary reasons for exclusion at this stage were insufficient methodological quality (n = 105) and incomplete results (n = 2). Ultimately, 20 studies met the full inclusion criteria and were included in this systematic review. The complete flow of the study selection process is illustrated in the PRISMA diagram (Figure 1).

### 3.2. Study Characteristics and Methodological Quality

The 20 included studies encompassed a variety of research designs: seven were clinical trials, seven were semi-experimental or applied designs, three were experimental, two were descriptive, and one was a quasi-experimental study. The participant populations included both males and females with ages ranging from 20 to 83 years. Sample sizes in the studies varied, ranging from 14 to 453 participants. The methodological quality of the included articles, assessed using the McMaster questionnaire, was varied. Eight studies were rated as “excellent” quality, seven as “very good” quality, and five as “good” quality. A detailed breakdown of the quality assessment scores for each study is presented in Table 2.

### 3.3. Intervention Characteristics

The interventions varied significantly across the studies. Core stability exercises were the most common intervention (20%). Other prominent interventions included combined exercise and massage protocols (15%); suspension exercises, yoga, and Pilates (15%); aquatic exercises (10%); and massage therapy alone (10%). The duration of interventions typically ranged from 2 to 8 weeks, with frequencies of two to four sessions per week. Session durations generally ranged from 20 to 60 min. While a few studies reported intensity using subjective measures like the Borg scale, none of the included studies reported standardized, objective information on the intensity of the interventions. Details of each intervention protocol are summarized in Table 3.

### 3.4. Main Outcomes

All 20 included studies reported positive outcomes, demonstrating that exercise-based rehabilitation is an effective strategy for managing lumbar DDD. Key findings from specific modalities include:

Aquatic exercises showed multiple benefits, including pain reduction, improved static and dynamic balance, increased range of motion, enhanced quality of life, and decreased disability.

Core stability exercises, comprising 20% of the studied interventions, showed comprehensive benefits. These benefits included reductions in pain intensity and functional disability, increases in range of motion and trunk flexibility, as well as improvements in core muscle endurance and performance.

In a direct comparison, one study found that suspension training was more effective than isolated core stability exercises in reducing pain (mean difference: −1.74; *p* < 0.001) and disability (mean difference: −8.21; *p* < 0.001).

Massage therapy, reported in 10% of the studies, significantly reduced pain but did not improve muscle endurance. Additionally, reflexology interventions were effective in alleviating pain, and some manual therapies demonstrated improvements in plantar pressure distribution. Other interventions, such as yoga and conventional rehabilitation programs, were also found to be effective. For example, yoga showed comparable effectiveness to conventional exercises in reducing pain and disability.

The McKenzie method also proved effective, with one trial demonstrating statistically significant advantages (*p* < 0.001) in improving spinal mobility and reducing pain compared to standard physiotherapy.

Comparative studies evaluated different modalities. For instance, one study comparing acupuncture and aquatic exercise found both interventions to be effective in improving pain, mobility, and quality of life, with no significant differences between them.

## 4. Discussion

The primary aim of this study was to systematically analyze the therapeutic effects of various rehabilitation exercises on lumbar DDD and to identify the most effective methods for preventing and alleviating back pain in affected individuals. The synthesis of the 20 included studies confirms the overall efficacy of exercise-based rehabilitation but also suggests a potential hierarchy of effectiveness among different modalities. Notably, the evidence indicates that dynamic and multi-component interventions may produce superior outcomes; for instance, a clinical trial by Mohebbi Rad et al. (2021) demonstrated that suspension training was more effective than isolated core stability exercises in reducing both pain (mean difference: −1.74) and disability (mean difference: −8.21), Aquatic therapy consistently demonstrated broad benefits across pain reduction, functional improvement, and balance [31]. This suggests that while most exercise-based approaches are beneficial, the choice of intervention can be tailored to target specific clinical goals.

The clinical benefits of hydrotherapy are multifaceted, primarily resulting from three key physiological mechanisms: substantial reduction of weight-bearing forces, enhanced sensation of lightness and fluid movement in the aquatic environment, and significant decrease in mechanical loading on articular structures. Additionally, the combined effects of hydrostatic weightlessness and the inherent massage-like properties of water immersion have been scientifically shown to produce measurable therapeutic outcomes, including notable reduction or complete resolution of muscle cramps and subsequent alleviation of disc-related muscular spasms [45,46].

Lotfi et al. [41] conducted a study investigating the effects of aquatic therapy, specifically examining how six weeks of supine water exercises impacted pain severity and disability levels in male patients with chronic low back pain due to lumbar disc herniation. Their results demonstrated statistically significant reductions in both pain intensity and disability within the experimental group, clearly indicating that aquatic exercise programs can effectively improve clinical outcomes in patients with lumbar pathologies.

Building on these findings, Mostaghel et al. [18] conducted a comparative study evaluating the effects of eight weeks of combined acupuncture and aquatic exercise therapy in patients with lumbar DDD. Their research assessed multiple outcomes, including pain levels, range of motion, and quality of life improvements. While both treatment modalities demonstrated comparable effectiveness in reducing pain, enhancing mobility, and improving quality of life, the study found no significant differences between the acupuncture and aquatic exercise groups, suggesting that either approach could be effectively incorporated into treatment protocols.

Further supporting evidence comes from Ezadi et al. [35], who implemented an eight-week program of selected aquatic exercises in a quasi-experimental study involving thirty female nurses with chronic low back pain from Sanandaj hospitals. Their results revealed significant between-group differences, with the intervention group demonstrating marked reductions in pain scores and clinically meaningful improvements in both static and dynamic balance measures. These positive outcomes were not observed in the control group, reinforcing aquatic exercise as an evidence-based intervention for enhancing motor function and alleviating pain in patients with lumbar DDD. Collectively, these studies position aquatic therapy as a particularly robust intervention due to its consistent, multi-domain benefits—including pain relief, balance enhancement, and improved quality of life—a breadth of effect not always achieved by more targeted, land-based exercises.

The collective results consistently identify stability exercises as an effective rehabilitation method, with documented improvements in both patient function and pain reduction. Clinical observations indicate altered movement patterns and impaired activation of the deep back muscles—the primary stabilizers of the lumbar region—in patients with chronic nonspecific low back pain [47,48].

These neuromuscular changes frequently contribute to pain development, muscular imbalances, and subsequent functional impairments, thereby reinforcing the therapeutic rationale for implementing stability exercises in both pain management and performance recovery [49].

Further supporting evidence is provided by Gandomi et al. [33], who specifically investigated plantar pressure distribution patterns, symmetry indices, and center of pressure fluctuations in female patients with discogenic low back pain. Their comparative analysis revealed significant differences in both pressure metrics and postural fluctuations between the disc herniation group and healthy controls, underscoring the critical role of core stability exercises in enhancing balance and motor control within this population.

Mohebbi Rad et al. [31] contributed to this body of evidence by directly comparing core stability exercises with suspension exercises in patients with lumbar disc herniation. Their findings showed that although both intervention groups achieved significant clinical improvements, the suspension exercise group experienced superior outcomes in terms of pain reduction and functional disability. These results were further supported by a follow-up study conducted by Mohebbi Rad et al. [32], which assessed electromyographic activity and reported markedly enhanced muscle activation patterns in the suspension exercise group, thereby providing physiological evidence for the effectiveness of suspension exercise interventions in improving physical function. This highlights a key theme in modern rehabilitation: while isolated core stability exercises form a foundational component, progression to more dynamic and challenging protocols, such as suspension training, may be necessary to maximize gains in pain relief and functional improvement.

Dzierżanowski et al. [4] investigated the effects of active exercises performed in low positions on lumbosacral function in patients with DDD. Their findings demonstrated that this specific exercise approach offers multiple clinical benefits: it significantly enhances the range of motion in the affected region, improves postural alignment, and reduces lower back pain. Moreover, these exercises led to measurable improvements in patients’ functional capacity for daily activities, highlighting their practical relevance in rehabilitation settings.

The accumulated body of evidence strongly supports the therapeutic value of aquatic exercise and complementary interventions for musculoskeletal disorders [42,43]. In light of these findings, we recommend integrating these modalities into comprehensive physiotherapy and multidisciplinary treatment programs for patients with spinal conditions. Emphasis should be placed on further research and the clinical implementation of these approaches. Several key findings support this recommendation: deep muscle training programs have demonstrated significant efficacy in alleviating back pain [50]; targeted exercise interventions effectively improve dysfunctional movement patterns [51]; and although some studies reported no significant differences between experimental and control groups in perceived pain or FMS scores, our review indicates that regular physical activity consistently leads to improved functional outcomes. Collectively, these findings reinforce the importance of exercise-based rehabilitation while emphasising the need for personalised treatment plans that account for individual patient characteristics.

The review also found that other modalities, while effective, generally demonstrated comparable rather than superior outcomes. For example, yoga was found to be as effective as conventional exercise, and manual therapies such as massage significantly reduced pain but did not improve muscle endurance. These findings suggest that for specific patient profiles or treatment goals, multiple viable options exist, with intervention selection potentially guided by patient preference and clinical presentation.

Despite the valuable insights provided by this systematic review, several limitations must be acknowledged. First, considerable heterogeneity was observed among the included studies in terms of intervention protocols—particularly regarding duration, intensity, and frequency—which complicates direct comparisons and precludes meta-analytic synthesis. Second, many trials enrolled fewer than 30 participants, a sample size widely recognized as underpowered for detecting moderate clinical effects, potentially limiting statistical power and the generalizability of findings [11,52]. Third, the current evidence base lacks robust longitudinal data, as most studies did not include long-term follow-up, thereby restricting the assessment of the durability of therapeutic effects. Consequently, while clinicians can confidently recommend these exercise interventions for short-term pain relief, their efficacy in preventing long-term recurrence remains uncertain.

Furthermore, our methodological quality appraisal revealed that the majority of studies were of high quality: 75% (15 out of 20) of the studies received an “excellent” or “very good” rating on the McMaster scale. However, the remaining variability suggests the potential for bias in outcome reporting. A closer examination of the quality appraisal (Table 2) revealed that the most prevalent methodological limitations among the included studies were the absence of sample size justification and inadequate reporting of participant withdrawal criteria.

Consequently, while the findings from studies with lower quality scores (e.g., Ghorbani et al. [38]; Kałużna et al. [36]) are included in our narrative synthesis, their contribution to the overall evidence should be interpreted with greater caution. Several studies also utilized quasi-experimental designs without random assignment, which increases susceptibility to selection bias, maturation effects, and other threats to internal validity [53,54]. Furthermore, the potential for publication bias—where studies reporting positive results are more likely to be published than those with null findings—cannot be excluded and may result in an overestimation of the true effect of these interventions. Lastly, although our search strategy was comprehensive, limiting the review to articles published in English and Persian may have led to the omission of relevant studies published in other languages.

Taken together, these limitations highlight an urgent need for future research involving larger, adequately powered randomised controlled trials (e.g., based on a priori power analyses to detect clinically meaningful effects, e.g., with sample sizes exceeding 100 participants per arm), standardised intervention protocols, and extended follow-up durations of at least 12 months post-intervention to better establish the long-term efficacy and generalizability of exercise-based rehabilitation for lumbar DDD.

## 5. Conclusions

This systematic review demonstrates that rehabilitation exercises, including hydrotherapy, core stability training, Pilates, and suspension exercises, effectively reduce pain and improve function in patients with lumbar DDD. The synthesized evidence suggests that when used as primary interventions, multi-component approaches like aquatic therapy and dynamic challenges like suspension training show particularly strong and broad benefits. Based on these findings, we recommend that clinicians consider prioritizing these modalities, especially for patients with significant functional deficits. While one study found no significant difference when combining aquatic therapy with another modality (acupuncture), this does not diminish its overall effectiveness as a core therapeutic strategy. However, it is crucial to acknowledge the limitations of the current evidence base. Therefore, to establish more definitive clinical guidelines, future research should prioritize larger, high-quality randomized controlled trials that employ standardized intervention protocols and include long-term outcome monitoring.

## Figures and Tables

**Figure 1 healthcare-13-01938-f001:**
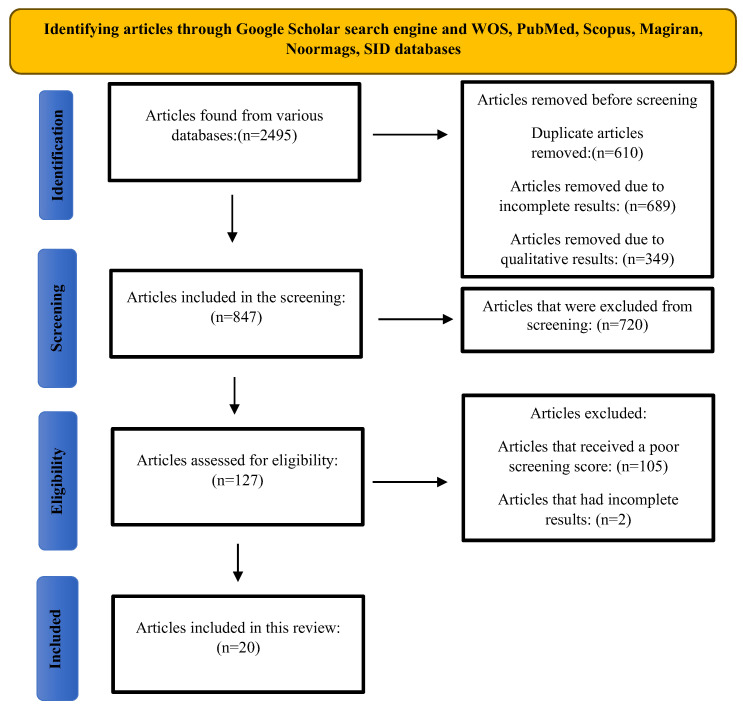
The process of searching, reviewing, and selecting articles based on the PRISMA guidelines.

**Table 1 healthcare-13-01938-t001:** Items and descriptions from the McMaster Critical Review Questionnaire for Quantitative Studies.

Item	Description
**1**	**Objective**: Was the purpose clearly stated?
**2**	**Literature review**: Was relevant background literature reviewed?
**3**	**Study design**: What type of study design was used?
**4**	**Sample size justification**: Was the sample size determination explained?
**5**	**Sample description**: Was the sample described in detail?
**6**	**Outcome reliability**: Were the outcome measures reliable?
**7**	**Outcome validity**: Were the outcome measures valid?
**8**	**Intervention details**: Was the intervention described in sufficient detail?
**9**	**Contamination**: Was contamination prevented?
**10**	**Co-intervention**: Were co-interventions avoided?
**11**	**Statistical significance**: Were statistically significant results reported?
**12**	**Statistical analysis**: Were the analytical methods appropriate?
**13**	**Clinical significance**: Was clinical significance reported?
**14**	**Withdrawal criteria**: Were withdrawal criteria reported?
**15**	**Conclusions**: Were conclusions appropriate given the methods and results?
**16**	**Clinical implications**: Were clinical implications of the findings discussed?
**17**	**Study limitations**: Were major study limitations or biases reported?

**Table 2 healthcare-13-01938-t002:** Quality assessment results based on McMaster’s critical questionnaire.

Author/Year	1	2	3	4	5	6	7	8	9	10	11	12	13	14	15	16	17	Total Score
Sobhani et al. (2023) [27]	1	1	1	1	1	1	1	1	1	0	1	1	1	1	1	1	1	16/16
Dehmordi & Fattahi (2022) [28]	1	1	1	1	0	1	1	1	1	1	1	0	1	1	1	0	0	13/16
Alhamashi et al. (2022) [29]	1	1	1	1	1	1	1	1	N/A	1	1	1	0	1	1	0	1	14/15
Mohebbi Rad et al. (2022) [30]	1	1	1	1	1	1	1	1	1	1	1	1	0	1	1	0	0	14/15
Mohebbi Rad et al. (2022) (2) [31]	1	1	1	1	1	1	1	1	1	1	1	1	1	1	1	1	0	16/16
Mohebbi Rad et al. (2022) (3) [32]	1	1	1	1	1	1	1	1	1	1	1	1	0	1	1	0	0	14/16
Gandomi et al. (2021) [33]	1	1	1	1	1	1	1	1	1	1	1	1	0	1	1	0	0	14/16
Hawrylak et al. (2021) [34]	1	1	1	0	1	1	1	1	1	1	1	1	1	1	1	1	1	16/16
Ezadi et al. (2021) [35]	1	1	1	0	1	1	0	1	1	1	1	1	1	1	0	0	0	12/16
Kałużna et al. (2019) [36]	1	1	1	0	0	1	1	0	1	1	1	1	1	0	1	0	0	11/16
Mostaghel et al. (2019) [18]	1	1	1	0	1	1	1	1	1	1	1	1	1	0	1	1	0	14/16
Lizis et al. (2017) [37]	1	1	1	0	1	1	1	1	1	1	1	1	1	1	1	1	1	16/16
Ghorbani et al. (2018) [38]	1	1	1	0	0	1	1	1	1	0	1	1	0	1	1	0	0	11/16
Ilbeigi et al. (2018) [39]	1	1	1	0	1	1	1	1	1	1	1	1	1	0	1	1	1	15/16
Teherán et al. (2016) [40]	1	1	1	0	1	1	1	1	1	1	1	1	0	1	1	1	1	15/16
Lotfi et al. (2015) [41]	1	1	1	0	1	1	1	0	1	1	1	1	1	0	1	0	0	12/16
Nazemzadeh et al. (2013) [42]	1	1	1	0	1	1	1	0	1	1	1	1	1	0	1	1	1	14/16
Dzierżanowski et al. (2013) [4]	1	1	1	0	1	1	1	1	N/A	1	1	1	1	0	1	1	1	14/15
Khanzadeh et al. (2012) [43]	1	1	1	0	1	1	1	1	1	1	1	0	1	1	1	1	0	14/16
Akbari & Rezaei (2011) [44]	1	1	1	0	1	1	1	1	1	1	1	1	0	1	0	0	1	13/16
Total Studies Meeting Criterion (n = 20)	20	20	20	7	17	20	19	17	18	18	20	18	13	14	18	10	9	-

1 = Criteria Fully Met, 0 = Criteria Not Fully Met. Quality Categories: Poor (0–8), Average (9–10), Good (11–12), Very Good (13–14), and Excellent (15–16). N/A: “Not applicable” was used for Item 9 (Contamination) only in single-group studies where the criterion was not relevant; for these studies, the denominator for the total score was adjusted to 15.

**Table 3 healthcare-13-01938-t003:** (**A**) Study Characteristics. (**B**) Intervention Protocols and Outcomes.

**(A)**								
**Authors**	**Study Objectives**	**Research Design**	**Population/Sample Size/Gender/Age**	**Intervention Duration**	**Frequency**	**Session Duration (min)**	**Intervention Intensity**	**Equipment**
Sobhani et al. (2024) [27]	Effectiveness of 8-week motor control retraining on chronic LBP in military males with lumbar disc herniation	Randomized Controlled Trial (RCT)	36 military males (intervention/control groups)/42.88 ± 7.96 years	8 weeks	3 sessions/week	45–60	NR	Electrotherapy
Dehmordi & Fattahi (2023) [28]	Effects of 6-week Pilates on disability and core metrics in women with lumbar disc herniation	Quasi-experimental (pre-post with control group)	30 women (15/15 groups)/55.6 ± 4.3 years	6 weeks	3 sessions/week	60	NR	Exercise equipment
Alhamashi et al. (2022) [29]	Short-term massage effects on plantar pressure and pain in women with discogenic LBP	Quasi-experimental (single-group pre-post)	14 women/35–45 years	5 sessions	Daily	15	NR	Force plate
Mohebbi Rad et al. (2021) [30]	Suspension training effects on EMG in men with chronic LBP from disc herniation	Non-Randomized Clinical Trial	22 men (12 suspension/10 control)/34.25 ± 8.81 years	8 weeks	3 sessions/week	30	Borg scale up to 6	EMG device
Mohebbi Rad et al. (2021) [31]	Comparison of core stability vs. suspension training on β-endorphins and pain	Non-Randomized Clinical Trial	32 men (10 core/12 suspension/10 control)/25–45 years	8 weeks	3 sessions/week	30	Borg scale: 5 (Core), 6 (Suspension)	BEURER scale, ZELL BIO kit
Mohebbi Rad et al. (2021) [32]	Effects 8-week core stability on abdominal muscle activation	Non-Randomized Clinical Trial	20 men (10/10 groups)/20–50 years	8 weeks	3 sessions/week	30	Borg scale up to 5	EMG device
Gandomi et al. (2021) [33]	Plantar pressure distribution patterns in women with discogenic LBP	Descriptive	34 women (17 LBP/17 healthy)/Mean 63 years	N/A	N/A	N/A	N/A	PT scan device
Hawrylak et al. (2021) [34]	McKenzie method efficacy for lumbar DDD	Experimental (pre-post with control group)	60 patients (40–59 years)	2 weeks (10 session)	Daily (weekdays)	30	NR	Digital inclinometer
Ezadi & Ghanizadeh Hesar (2021) [35]	Aquatic exercises’ effects on pain/balance in nurses with chronic LBP	Quasi-experimental (pre-post with control group)	30 female nurses (15/15 groups)/25–40 years	8 weeks	3 sessions/week	NR	NR	-
Kałużna et al. (2019) [36]	Stabilization exercises for lumbar DDD	Experimental (pre-post with control group)	30 patients (16F/14M)/25–68 years	14 days	NR	NR	NR	-
Mostaghel et al. (2019) [18]	Acupuncture + aquatic exercise for DDD	Quasi-experimental (comparative design)	24 women (12/12 groups)	8 weeks	3 sessions/week	45–60	NR	-
Latafatkar et al. (2018) [38]	Stability exercises vs. reflexology for chronic LBP	Quasi-experimental (multi-group design)	47 women (4 groups)/30–35 years	8 weeks	3 sessions/week	15–30	NR	-
Ilbeigi et al. (2018) [39]	Kinesiotaping vs. foot reflexology for non-specific LBP	Quasi-experimental (multi-group design)	30 men (3 groups)/20–40 years	6 weeks	3 sessions/week	30	NR	-
Lizis et al. (2017) [37]	Kaltenborn-Evjenth Orthopaedic Manual Therapy (KEOMT) vs. kinesiotherapy for DDD	Randomized Controlled Trial (RCT)	80 patients (40–70 years)	5 weeks (10 treatments)	2 sessions/week	30 (KEOMT), 45 (Kinesiotherapy)	NR	Kinesiotape
Teherán et al. (2016) [40]	Alternative therapies for lumbar disc disease	Descriptive, Retrospective Study	453 patients/Mean 57 years	Variable (>6 months)	N/A	N/A	N/A	-
Lotfi et al. (2016) [41]	Supine aquatic exercises for men with discogenic LBP	Quasi-experimental (pre-post with control group)	24 men (12/12 groups)/Mean 38.83 ± 5.78 years	6 weeks (24 sessions)	4 sessions/week	45–60	NR	-
Nazemzadeh et al. (2013) [42]	Foot reflexology physiological effects	Randomized Controlled Trial (RCT)	150 men (3 groups)/45.8 ± 13.39 years	3 weeks	1 session/week	30	NR	Sphygmomanometer
Dzierżanowski et al. (2013) [4]	Low-position exercises for lumbar-sacral DDD	Experimental (single-group pre-post)	20 patients (17F/3M)/24–73 years	2 weeks (10 days)	Daily (weekdays)	20	NR	-
Khanzadeh et al. (2012) [43]	Combined exercise + massage protocol	Quasi-experimental (pre-post with control group)	30 men (15/15 groups)/41.61 ± 4.98 years	8 weeks	3 sessions/week	60	NR	-
Akbari & Rezaei (2012) [44]	Yoga effects on flexibility/pain in chronic disc herniation	Double-Blind Randomized Controlled Trial (RCT)	28 women (14/14 groups)/30–40 years	8 weeks (16 sessions)	2 sessions/week	45	NR	Flexible ruler
(**B**)								
**Authors**	**Research** **protocol**		**Results**			**Conclusions**		
Sobhani et al. (2024) [27]	Functional disability questionnaire, VAS pain scale		Significant improvements in pain intensity (*p* < 0.05), functional disability, and range of motion.			Motor control retraining effectively improves pain, disability, and ROM, and may enhance QoL in military personnel.		
Dehmordi & Fattahi (2023) [28]	Oswestry Disability Index, Schober test, Pilates tests		6 weeks of Pilates significantly improved disability, trunk flexibility, and core muscle strength/endurance.			Pilates rehabilitation is a recommended intervention for symptom management in patients with lumbar disc herniation.		
Alhamashi et al. (2022) [29]	Plantar pressure assessment and pain intensity evaluation		Significant pain reduction was observed after 5 massage sessions. Plantar pressure improved significantly after 5 sessions, but not after a single session.			While a single massage session can reduce pain, five sessions may be needed to influence compensatory mechanisms related to plantar pressure.		
Mohebbi Rad et al. (2021) [30]	EMG assessment and straight leg lowering test		Suspension group showed significant improvement in EMG activity of four muscles and strength test performance compared to controls.			Suspension training enhances core muscle activation and is a valuable component of rehabilitation programs.		
Mohebbi Rad et al. (2021) [31]	Core stability and suspension training protocols		Both exercise types improved β-endorphin, pain, and disability. Suspension training demonstrated superior pain and disability reduction compared to core stability.			Suspension training appears more effective for pain and disability management in lumbar disc herniation patients.		
Mohebbi Rad et al. (2021) [32]	EMG activity, abdominal strength, and flexibility tests		8 weeks of core stability training led to significant improvements in muscle activation, strength, and flexibility.			Core stability exercises lead to notable improvements in trunk muscle function and should be included in rehabilitation.		
Gandomi et al. (2021) [33]	Center of pressure and symmetry index assessment		Disc herniation patients showed significant increases in sway velocity and path length, and reduced symmetry, compared to controls.			Neuromuscular training for spinal stabilizers may help reduce subsequent injury risk in this population.		
Hawrylak et al. (2021) [34]	Conservative McKenzie physiotherapy method		Both the McKenzie and standard physiotherapy groups improved, but enhanced outcomes were observed in the McKenzie group.			The McKenzie method is a clinically useful intervention for lumbar DDD management.		
Ezadi & Ghanizadeh Hesar (2021) [35]	Stork test, Berg test, Quebec Disability Scale		The aquatic exercise group showed significant pain reduction and balance improvement compared to the control group.			Aquatic exercise protocols focusing on core muscle strengthening should be emphasized in therapeutic programs.		
Kałużna et al. (2019) [36]	Functional Movement Screen (FMS), VAS pain scale		Deep muscle stabilization exercises significantly reduced lumbar pain and positively affected global movement patterns.			Stabilization exercises positively impact pain reduction and functional status in patients with lumbar DDD.		
Mostaghel et al. (2019) [18]	VAS, Straight Leg Raise test, SF-36 questionnaire		8 weeks of combined acupuncture and aquatic exercise led to significant improvements in pain, ROM, and quality of life.			Both acupuncture and aquatic exercise are effective for pain reduction, mobility improvement, and quality of life.		
Latafatkar et al. (2018) [38]	McGill functional tests, VAS pain scale		Stability exercises (alone and combined) improved muscle endurance and pain with durable effects. Reflexology massage only showed positive effects on pain reduction.			Due to their durable effects, core stability exercises should be prioritized in chronic LBP treatment protocols.		
Ilbeigi et al. (2018) [39]	VAS pain scale, Oswestry Disability Index		Both kinesiotaping and foot reflexology reduced pain and disability. Kinesiotaping showed significantly greater pain reduction than reflexology.			While both methods are effective, kinesiotaping may be preferable for achieving greater pain relief.		
Lizis et al. (2017) [37]	SF-36 quality of life form, VAS pain scale		After 5 weeks, KEOMT showed statistically significant advantages over kinesiotherapy for all QoL domains and pain.			KEOMT is more effective than kinesiotherapy for improving quality of life and pain in lumbar DDD patients.		
Teherán et al. (2016) [40]	Alternative therapy evaluation		Alternative therapies demonstrated a 50–60-point pain reduction, sustained for over 6 months.			Alternative therapies show significant impact on chronic pain in patients unresponsive to conventional treatments.		
Lotfi et al. (2016) [41]	Aquatic movement therapy, Student’s t-test		The aquatic exercise group showed significant reductions in pain intensity and greater disability reduction compared to the control group.			Supine aquatic exercises can effectively reduce pain and functional disability in men with chronic discogenic LBP.		
Nazemzadeh et al. (2013) [42]	Physiological measures (respiratory rate, pulse, BP)		Post-intervention, all three groups (intervention, placebo, control) showed reduced respiratory rate and blood pressure.			Foot reflexology may effectively improve physiological indicators in LBP patients, suggesting clinical utility.		
Dzierżanowski et al. (2013) [4]	Schober test, finger-to-floor test, seated spinal rotation		90% of patients reported improved lateral spinal flexion after 2 weeks of rehabilitation.			Active low-position exercises significantly improve range of motion, postural alignment, and pain reduction.		
Khanzadeh et al. (2012) [43]	Combined exercise and massage protocol		The experimental group showed significant improvements in physical function and pain scores after 8 weeks.			A combined exercise and massage protocol is an effective approach for improving function and reducing pain.		
Akbari & Rezaei (2012) [44]	VAS, Oswestry Disability Index, Beck Depression Inventory		Both yoga and conventional exercises significantly reduced pain, disability, depression, and lordosis, and improved ROM. No significant differences were found between the two interventions.			Yoga and conventional exercises show comparable effectiveness for managing symptoms in women with chronic lumbar disc herniation.		

Notes: N/A: Not applicable, NR: Not reported.

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
