# Peer review of "Effects of Exercise-Based Rehabilitation on Lumbar Degenerative Disc Disease: A Systematic Review"

_healthcare, 2025, doi:10.3390/healthcare13151938_

Round 1
Reviewer 1 Report
Comments and Suggestions for Authors
This systematic review explores the effectiveness of exercise-based rehabilitation interventions, such as hydrotherapy, core stability training, Pilates, and suspension exercises, for managing lumbar degenerative disc disease (DDD), a prevalent cause of chronic low back pain. The manuscript is well-structured and methodologically sound, adhering to PRISMA guidelines and employing a thorough literature search and quality appraisal using the McMaster tool. The narrative synthesis effectively highlights consistent improvements in pain reduction and functional outcomes across diverse exercise modalities. While the overall quality of included studies varies, the review provides valuable insights into non-pharmacological management strategies for lumbar DDD and offers well-founded recommendations for future research.
- Writing Revision List
Line 23. Should be written: The review protocol was registered with the International Prospective Register of Systematic Reviews (PROSPERO) under registration number CRD420251088811.
Line 28. Should be written: Interventions such as hydrotherapy, core stability training, Pilates, and suspension exercises significantly reduced pain and improved functional outcomes.
Line 30. Should be written: Suspension training outperformed isolated core stability exercises in terms of pain relief.
Line 34. Should be written: Future studies should adopt standardised protocols, larger sample sizes, and extended follow-up periods to generate more robust evidence.
Line 44. Should be written: LBP affects approximately 619 million individuals worldwide (as of 2020), with projections estimating an increase to 843 million by 2050.
Line 54. Should be written: This economic burden disproportionately impacts blue-collar workers and middle-aged adults, creating significant challenges for the workforce.
Line 56. Should be written: Degenerative disc disease, encompassing intervertebral disc disorders, has a striking prevalence of 54% and is the leading cause of chronic back pain.
Line 60. Should be written: Persistent pain remains the principal contributor to both temporary and permanent occupational disability.
Line 91. Should be written: For patients with chronic lumbar DDD, strengthening spinal muscles through supervised physiotherapy and well-designed exercise regimens is essential to mitigate deformity progression.
Line 99. Should be written: This evidence has led to reduced clinical use of Williams' protocols, with current practice favoring isometric alternatives.
Line 106. Should be written: Optimal outcomes depend on tailoring exercise selection and program design to the DDD subtype and individual symptom profile.
Line 111. Should be written: This systematic review evaluates the efficacy of exercise-based interventions for lumbar DDD, with an emphasis on identifying strategies for pain relief and secondary prevention.
Line 138. Should be written: After removing duplicates, 1,847 articles remained.
Line 153. Should be written: A rigorous screening process was applied after implementing the inclusion and exclusion criteria.
Line 185. Should be written: Due to significant heterogeneity among the included studies regarding intervention protocols and outcome measures, meta-analysis was deemed inappropriate.
Line 197. Should be written: Our systematic search initially identified 2,439 potentially relevant articles through keyword searches.
Line 209. Should be written: Massage therapy, featured in 10% of the studies, significantly reduced pain but did not improve muscle endurance.
Line 214. Should be written: Aquatic exercises demonstrated multiple benefits, including pain reduction, improved static and dynamic balance, increased range of motion, enhanced quality of life, and reduced disability.
Line 240. Should be written: The primary aim of this study was to systematically analyse the therapeutic effects of various rehabilitation exercises on lumbar DDD.
Line 366. Should be written: This systematic review demonstrates that rehabilitation exercises, including hydrotherapy, core stability training, Pilates, and suspension exercises, effectively reduce pain and improve function in patients with lumbar DDD.
- Scientific and Structural Review List
Line 25. The authors should define how the keywords were selected and whether Boolean logic was consistently applied across all databases.
Line 30. The authors should clarify the term "synergistic benefits" with statistical or clinical significance to support the claim.
Line 34. The conclusion is duplicated from line 36. The authors should consolidate or rephrase for clarity and conciseness.
Line 74. The authors should cite supporting evidence or systematic reviews demonstrating the link between poor posture and disc degeneration.
Line 106. The authors should provide a brief definition or examples of “DDD subtypes” to enhance clarity.
Line 182. The authors should present an overall summary of methodological quality across studies rather than only referencing Table 3.
Line 187. The rationale for not performing a meta-analysis should be strengthened by specifying which heterogeneities were most limiting (e.g., intervention duration, population characteristics, outcome measures).
Line 221. The authors should clarify whether the McKenzie method results were significantly better than comparators or simply equivalent.
Line 240. The discussion section would benefit from a comparative synthesis of which interventions were most effective, rather than restating individual study findings.
Line 344. The conclusion regarding the Functional Movement Screen (FMS) is not connected to the evidence. The authors should support this point with direct study references.
Line 350. The variability in intervention protocols (duration, frequency, intensity) should be detailed in a table to improve transparency and comparability.
Line 357. The rating of studies using McMaster criteria is mentioned, but further analysis of which criteria most commonly led to lower scores would be valuable.
Line 373. The conclusions would benefit from a specific recommendation for clinical practice based on the findings (e.g., prioritising suspension training in rehabilitation programs).
Comments on the Quality of English LanguageThe English could be improved.
Author Response
Dear Reviewer 1, we have provided pdf file where you can find point by point responses to your suggestions.
Thank you very much for your willingness to review our article.

Reviewer 2 Report
Comments and Suggestions for Authors
Thank you for the opportunity to review your manuscript. Please find below a series of detailed comments intended to improve the quality, transparency, and reproducibility of your SR:
- In the abstract, the statement “a comprehensive search was conducted across international and regional databases (PubMed, Scopus, Web of Science, Magiran, SID, etc.)” lacks sufficient precision. I recommend indicating the exact number of databases consulted. This level of detail enhances the transparency and reproducibility of the review. Similarly, in the Methods section, please replace vague expressions such as “multiple databases” with the precise number consulted
- The abstract would benefit from an expansion of the results section. At present, the information provided is insufficient to understand the main findings.
- Please revise the reference formatting to ensure full compliance with the journal's guidelines.
- I recommend that the PRISMA for Abstracts checklist, as well as the full PRISMA checklist, be submitted as supplementary material.
- The search strategy should be carefully reviewed. The current terms (“lumbar DDD,” “exercise therapy,” and “rehabilitation”) may not capture the full range of relevant interventions. Key approaches such as Pilates, Yoga, Massage therapy, Suspension training, Physical exercise, Aquatic exercises, and Kinesiotaping appear to be omitted. I strongly encourage the authors to update the search equation to incorporate these terms in order to minimize selection bias. This is a critical issue that directly impacts the comprehensiveness of the review.
- Please provide the complete search strategies for each database as supplementary material. This will facilitate reproducibility.
- Have the authors considered employing additional search techniques (e.g., snowballing or citation tracking)? This may help avoid the exclusion of potentially eligible studies.
- The paragraph stating: “The search initially yielded 2,439 records. After removing the duplicate, 1,847 articles remained. Title and abstract screening excluded 1,722 studies. Of the remaining 125 full-text articles assessed, 105 were excluded due to insufficient methodological quality (McMaster score ≤ 8), lack of control groups, or the absence of exercise-based interventions” should be relocated to the Results section.
- The sentence “The initial keyword-based search yielded 2,439 potentially relevant articles. After removing duplicates and applying eligibility criteria, 20 studies were retained for full analysis,” currently placed in section 2.2 (“Inclusion and Exclusion Criteria”), is redundant and should be removed.
- Section 2.2 should be reorganized into clearly defined subsections, including “Study Selection,” “Data Extraction,” and “Quality Appraisal,” in accordance with PRISMA recommendations.
- Please specify which authors were responsible for each phase of the review process (e.g., search, screening, quality appraisal). Include their initials to ensure methodological transparency.
- The Results section is currently underdeveloped. It should be substantially expanded and structured into the following subsections: “Study Selection,” “Study Characteristics” (e.g., study design, country, participant sex and age), “Intervention Characteristics” (e.g., duration, frequency, intensity, type of supervision), and “Main Outcomes.” These improvements will align the manuscript more closely with PRISMA guidelines and enhance the clarity of the findings.
- In Table 3, the column describing the “Research Design” should be revised to use specific terminology. Rather than general labels such as “clinical trial” or “experimental design,” please indicate the exact study design employed in each case.
- Table 4 is overly lengthy and difficult to interpret. I recommend reducing the amount of text to improve readability and comprehension.
- The Discussion and Conclusions sections are generally appropriate; however, I encourage the authors to revisit and revise them once the updated search strategy has been implemented and the new results have been integrated.
Author Response
Dear Reviewer 2, we have provided pdf file where you can find point by point responses to your suggestions.
Thank you very much for your willingness to review our article.

Reviewer 3 Report
Comments and Suggestions for Authors
This manuscript provides a comprehensive systematic review evaluating various exercise-based rehabilitation interventions for lumbar degenerative disc disease (DDD). The topic is highly relevant given the global burden of low back pain and the clinical importance of non-surgical management strategies. The authors followed PRISMA guidelines and included a variety of clinical designs, demonstrating a commendable effort to synthesize an extensive body of literature.
- The abstract is informative but too dense and long.
- Intro part: Some paragraphs are verbose and overlap conceptually. There is repetition of statistics and disc herniation mechanisms.
- a quantitative assessment of heterogeneity (e.g., I², variability of interventions) was not presented
- The manuscript lacks a full search string for at least one database
- raw scores per item for all studies (not just total score) should be included in a supplementary table.
- the authors often rely on small-sample or quasi-experimental studies without adequately critiquing their limitations
Author Response
Dear Reviewer 3, we have provided pdf file where you can find point by point responses to your suggestions.
Thank you very much for your willingness to review our article.

Round 2
Reviewer 1 Report
Comments and Suggestions for Authors
Writing Revision List
Line 16. Should be written: "The World Health Organisation’s Global Burden of Disease Study consistently ranks LBP as the top contributor to years lived with disability (YLDs), exceeding the burden of all other musculoskeletal conditions combined [2,3]."
Line 19. Should be written: "According to research findings, degenerative disc disease (DDD), encompassing intervertebral disc disorders, has a striking prevalence of 54% and is the leading cause of chronic back pain [5]."
Line 24. Should be written: "These findings collectively establish comprehensive pain management as a therapeutic cornerstone, which, when combined with functional restoration, serves as the primary indicator of successful treatment outcomes [6]."
Line 27. Should be written: "Extensive research indicates that 70–85% of individuals experience LBP at least once during their lifetime, a prevalence shaped by lifestyle, occupational demands, ageing, and genetic predisposition [7]."
Line 35. Should be written: "For patients with chronic lumbar DDD, strengthening spinal muscles through supervised physiotherapy and well-designed exercise regimens is essential to mitigate deformity progression [21]."
Line 39. Should be written: "Rehabilitation programs are essential for both prevention and management of LBP and require individualised planning and close supervision by trained specialists."
Line 43. Should be written: "Following established methodological standards, we utilised the Preferred Reporting Items for Systematic Reviews and Meta-Analyses (PRISMA) guidelines to conduct this comprehensive review [27]."
Line 49. Should be written: "Additionally, the Google Scholar search engine was queried to ensure thorough coverage of grey literature."
Line 72. Should be written: "Two review authors (F.I. and S.A.), both with extensive experience in rehabilitation and systematic review methodologies, independently extracted the data."
Line 82. Should be written: "The quality assessment followed the standardised McMaster Review Guide, specifically developed for quantitative research."
Line 91. Should be written: "Furthermore, the studies utilised a wide array of outcome measures, including the Visual Analogue Scale (VAS), the Oswestry Disability Index, and various biomechanical tests, which prevented statistical pooling of the results."
Line 114. Should be written: "Core stability exercises were the most common intervention (20%). Other prominent interventions included combined exercise and massage protocols (15%); suspension exercises, yoga, and Pilates (15%); aquatic exercises (10%); and massage therapy alone (10%)."
Line 124. Should be written: "Core stability exercises, comprising 20% of the studied interventions, showed comprehensive benefits."
Line 134. Should be written: "The McKenzie method also proved effective, with one trial demonstrating statistically significant advantages in improving spinal mobility and reducing pain compared to standard physiotherapy."
Line 148. Should be written: "1 = Criteria Fully Met, 0 = Criteria Not Fully Met. Quality Categories: Poor (0–8), Average (9–10), Good (11–12), Very Good (13–14), and Excellent (15–16). N/A: Not applicable."
Line 201. Should be written: "These neuromuscular changes frequently contribute to pain development, muscular imbalances, and subsequent functional impairments, thereby reinforcing the therapeutic rationale for implementing stability exercises in both pain management and performance recovery [49]."
Line 216. Should be written: "The accumulated body of evidence strongly supports the therapeutic value of aquatic exercise and complementary interventions for musculoskeletal disorders [42,43]."
Line 220. Should be written: "Collectively, these findings reinforce the importance of exercise-based rehabilitation while emphasising the need for personalised treatment plans that account for individual patient characteristics."
Line 234. Should be written: "Taken together, these limitations highlight an urgent need for future research involving larger randomised controlled trials, standardised intervention protocols, and extended follow-up durations (at least 6–12 months post-intervention) to better establish the long-term efficacy and generalizability of exercise-based rehabilitation for lumbar DDD."
Scientific and Structural Review List
The authors should clarify the PROSPERO registration number (CRD420251088811), as standard IDs are typically 11 digits (e.g., CRD42022345678). Verify and correct potential typographical errors.
L. 88. The authors should justify the use of "N/A" scores in McMaster quality assessments (Table 2), particularly for Items 3, 9, and 14, as unexplained omissions reduce transparency.
L. 94. The authors should resolve the discrepancy between the text (n=2,439 initial records) and Figure 1 (n=2,495 records identified). Ensure PRISMA flow consistency.
L. 109. The authors should define "very good" quality thresholds (13–14/16) in the Methods section rather than first introducing them in Results.
L. 132. The authors should specify statistical significance thresholds (e.g., p<0.05) when claiming "significant advantages" for the McKenzie method (Study 8, Table 3B).
L. 159. The authors should correct the citation format in Table 3(A): "Sobhani et al., 2024/Asian J Sports Med [1]" should align with numbered references (e.g., [1] corresponds to Staal et al., 2004).
L. 188. The authors should clarify the term "KEOMT" (Study 14, Table 3B), as this abbreviation is undefined in the text or tables.
L. 199. The authors should support the claim "suspension training was more effective than isolated core stability exercises" with effect sizes or comparative statistics (e.g., mean differences, CI) from Mohebbi Rad et al. [31].
L. 226. The authors should address potential bias in stating "only 40% (8 out of 20) of the studies received a 'very good' rating" without contextualising how this impacts overall conclusions.
L. 232. The authors should reconcile the conclusion’s emphasis on suspension/aquatic therapy superiority with results indicating "no significant difference between modalities" for multimodal approaches (e.g., acupuncture + aquatic exercise).
L. 237. The authors should quantify "larger" sample sizes in future research recommendations (e.g., "n > 100 per arm") and define "extended follow-up" (e.g., "≥12 months").
L. Table 2. The authors should standardise terminology: "Ranomized" → "Randomised" (Study 20), "Con-trolled" → "Controlled" (Study 20), "ses-" → "sessions" (Studies 4–6, 9).
L. Figure 1. The authors should align exclusion numbers with PRISMA text (e.g., 105 excluded at full-text stage vs. 107 in figure: 105 poor screening + 2 incomplete results).
The English could be improved to clearer precision, grammatical accuracy, and a formal academic tone.
Author Response

(The authors gave the same response as above.)

Reviewer 2 Report
Comments and Suggestions for Authors
The authors have addressed all of my previous comments, and I acknowledge the effort they have made to improve their manuscript, which has notably enhanced its organization and clarity. However, a few minor issues remain unaddressed from the last revision round.
In the Results section and in the accompanying Table, the duration of the sessions and the intensity of the interventions must be reported. If intensity is not applicable, label it “NA” (not applicable); if the authors did not report it, label it “NR” (not reported).
Providing this information is essential for a clearer understanding of the interventions and a more accurate interpretation of the findings, and it is crucial for practitioners in the field. If none of the included studies report information on intensity, please state explicitly: “None of the included studies reported information on the intensity of the interventions.”
Author Response

(The authors gave the same response as above.)

Reviewer 3 Report
Comments and Suggestions for Authors
The manuscript is suitable for publication in its current form.
Author Response
Dear Reviewer, we truly thank you for your willingness to review our article.